# Machine Learning and Remote Sensing Application for Extreme Climate Evaluation: Example of Flood Susceptibility in the Hue Province, Central Vietnam Region

Minh Cuong Ha [1], Phuong Lan Vu [2], Huu Duy Nguyen [2], Tich Phuc Hoang [1], Dinh Duc Dang [3], Thi Bao Hoa Dinh [2], Gheorghe Şerban [4,*], Ioan Rus [4,*] and Petre Brețcan [5]

[1] School of Aerospace Engineering (SAE), VNU University of Engineering and Technology (UET), Hanoi 100000, Vietnam; cuonghm@vnu.edu.vn (M.C.H.); phucth@fimo.edu.vn (T.P.H.)
[2] Faculty of Geography, VNU Hanoi University of Science (HUS), Hanoi 100000, Vietnam; vuphuonglan@hus.edu.vn (P.L.V.); nguyenhuuduy@hus.edu.vn (H.D.N.); dinhthibaohoa@hus.edu.vn (T.B.H.D.)
[3] Center for Environmental Fluid Dynamics, VNU Hanoi University of Science (HUS), Hanoi 100000, Vietnam; dangduc@hus.edu.vn
[4] Faculty of Geography, Babeș-Bolyai University, 400084 Cluj-Napoca, Romania
[5] Department of Geography, Faculty of Humanities, Valahia University, Lt. Stancu Ion 35, 130105 Targoviste, Romania; petrebretcan@yahoo.com
[*] Correspondence: gheorghe.serban@ubbcluj.ro (G.Ş.); ioan.rus@ubbcluj.ro (I.R.)

**Abstract:** Floods are the most frequent natural hazard globally and incidences have been increasing in recent years as a result of human activity and global warming, making significant impacts on people's livelihoods and wider socio-economic activities. In terms of the management of the environment and water resources, precise identification is required of areas susceptible to flooding to support planners in implementing effective prevention strategies. The objective of this study is to develop a novel hybrid approach based on Bald Eagle Search (BES), Support Vector Machine (SVM), Random Forest (RF), Bagging (BA) and Multi-Layer Perceptron (MLP) to generate a flood susceptibility map in Thua Thien Hue province, Vietnam. In total, 1621 flood points and 14 predictor variables were used in this study. These data were divided into 60% for model training, 20% for model validation and 20% for testing. In addition, various statistical indices were used to evaluate the performance of the model, such as Root Mean Square Error (RMSE), Receiver Operation Characteristics (ROC), and Mean Absolute Error (MAE). The results show that BES, for the first time, successfully improved the performance of individual models in building a flood susceptibility map in Thua Thien Hue, Vietnam, namely SVM, RF, BA and MLP, with high accuracy (AUC > 0.9). Among the models proposed, BA-BES was most effective with AUC = 0.998, followed by RF-BES (AUC = 0.998), MLP-BES (AUC = 0.998), and SVM-BES (AUC = 0.99). The findings of this research can support the decisions of local and regional authorities in Vietnam and other countries regarding the construction of appropriate strategies to reduce damage to property and human life, particularly in the context of climate change.

**Keywords:** flood; BES; SVM; Hue; Vietnam

## 1. Introduction

Due to the number of people affected, globally floods have the most significant impact of all natural hazards each year. They represent more than half of the world's catastrophic events [1–3]. According to the EM-DAT database, from 2000 to 2018 more than 2900 floods were recorded, causing more than 295,000 deaths, affecting more than 1.5 billion people, and causing overall economic losses of nearly 503 billion USD. Asia is the region most affected by flooding, causing significant damage to the economy and human life [4–8]. During 1900–2016, countries in the region were affected by 3620 floods and typhoons,

resulting in over 683 billion USD of damage [9]. Due to its geographic location and dense hydrological network, Vietnam is considered to be one of the countries most affected by flooding [10]. It has been ranked among the eight countries most affected by extreme weather conditions. For example, the 1999 flood caused a loss to the economy of over USD 300 million, 547 deaths, and over 630,000 houses damaged [11]. Therefore, it is necessary to develop an appropriate strategy to reduce this considerable impact on people's livelihoods and wider socio-economic activities.

Urbanization, continued population growth, deforestation and climate change predict an increase in flood risk, because urbanization impinges on agricultural areas close to cities, and urban heat islands are formed [12–15]. At the same time, the use of structural measures such as dikes and dams has proved inefficient and costly in the long term [16]. Therefore, prediction of areas of flood susceptibility is considered to be one of the most important strategies to allow planners to optimize the management and sustainable planning of land use.

Several methods have been developed in the literature, including physically based hydrologic models such as MIKE FLOOD [17,18], Hydrologic Engineering Center's River Analysis System (HEC-RAS) [19,20], and Soil Water Assessment Tool (SWAT) [21,22]. These methods have proven effective in flood hazard assessment. However, the use of these models has important limitations when applied to large regions, particularly in regions with limited data, because they require detailed in-situ data such as river cross-sectional data and long-term meteorological and hydrological data [23,24]. With the growth of the relevant technology and open data, remote sensing is considered to be an effective tool for monitoring flooding [25]. Regional data has been collected from satellites using the remote sensing method combined with Synthetic Aperture Radar (SAR) data [26,27] or optical images [28,29] to detect flood zones. However, floods are usually short-lived and occur during periods of inclement weather; therefore, in a few cases, the sensors do not register the correct flood time and are often affected by cloudiness [5]. Additionally, although remote sensing and GIS can analyze the spatial distribution of flooding, they have critical limitations in representing the latent impacts and causes of flooding [30].

To accommodate these limitations, the global scientific community has integrated spatial analysis and data-driven models to identify large regions of flood susceptibility with high precision, even for regions with limited data. Previous studies have used traditional models, including bivariate statistical models [31], frequency ratio [32], weights-of-evidence [33], and fuzzy weight of evidence [34]. However, these models are mainly linear structures and fail to address the complex and non-linear structure of flooding events. Currently, there is a new trend of employing artificial intelligence in combination with remote sensing and GIS to better visualize and assess the effects of flooding, such as support vector machine (SVM) [35,36], artificial neural networks (ANN) [37], Kernel Logistic Regression (KLR) [38,39], decision trees (DT) [40,41], Random Forest (RF) [41,42], Adaboost (ADB) [43,44], and Bagging [45,46]. These have several advantages over traditional models: (i) They can predict complex non-linear structures of flood events with high accuracy, even in data-bound regions and (ii) they can easily combine with spatial data obtained from satellite images. However, the flood susceptibility model encounters several obstacles, such as selecting the best modeling methods from a wide range, and each approach has obtained different results. In addition, each approach for predicting flood susceptibility has several disadvantages [25]. Therefore, the search for new optimization algorithms to improve the ability of prediction is necessary because hybrid models limit the weak points of individual models by combining two or more models [47]. Various optimization algorithms have been developed in the literature, divided into two groups: Evolutionary Algorithms (EA) and Swarm intelligence (SI). EA is an intelligent computing technique applied widely to solve several problems such as combinatorial and non-linear optimization [48]. This technique can easily solve different problems by integrating previous information into the evolutionary research process to effectively exploit a state space of possible solutions [49]. SI solves the problem by simulating animal behaviors, including exploration and exploitation [49].

SI has the advantage in solving computational and combinatorial problems thanks to its ability to find the best position in the swarm [50]. However, the global issue of optimization and preservation of optimization until the end of the research is seen as one of the essential challenges in solving several problems [51]. There is no universal guide for selecting the best method for flood susceptibility. Therefore, it is necessary to exploit novel algorithms to complete these gaps. This study proposes a new nature-inspired technique which combines the advantages of SI and EA, namely Bald Eagle Search (BES), which simulates the behavior of bald eagles when searching for food. This process is carried out in three processes: first, bald eagles identify an area within which to search for food; second, they search for food in that area; third, they attack their prey [49,52]. The algorithm is widely applied in various fields such as ecology, energy, and economics; however, according to the best understanding of the authors, it has not yet been explored in the field of earth sciences to predict flood susceptibility prediction.

The objective of this research is the development of advanced new models based on machine learning and remote sensing to assess flood susceptibility in Thua Thien Hue province, Vietnam. This study differs from previous studies in that it represents the first use of BES to optimize four individual models: SVM, BA, RF, and MLP. The hypothesis tested in this study is that BES is able to optimize base models to perform better than regular base models would. Although these models are applied to Vietnam, results can be used to inform work in other parts of the world, even in regions with limited data. The findings of this study can be seen as an important aid to decision-makers by highlighting areas that have high susceptibility to flooding and therefore are not suitable for development, helping to minimize the impacts of floods on the economy and human lives.

## 2. Materials and Methods

Thua Thien Hue province is located in the central region of Vietnam (Figure 1). It covers an area of 5055 km$^2$ and has 1283 million inhabitants. The topography of the province is very complex, varying from 0 to 1774 m, and is divided into three distinct areas: mountains, plains, and coastal lagoons. Of these, mountains constitute approximately 85% of the total area, plains 10%, and lagoons 5%. The province also has a dense hydrological network of 0.3–2.5 km/km$^2$ with three main rivers: the Perfume River, the Bo River, and the Ong Lau River.

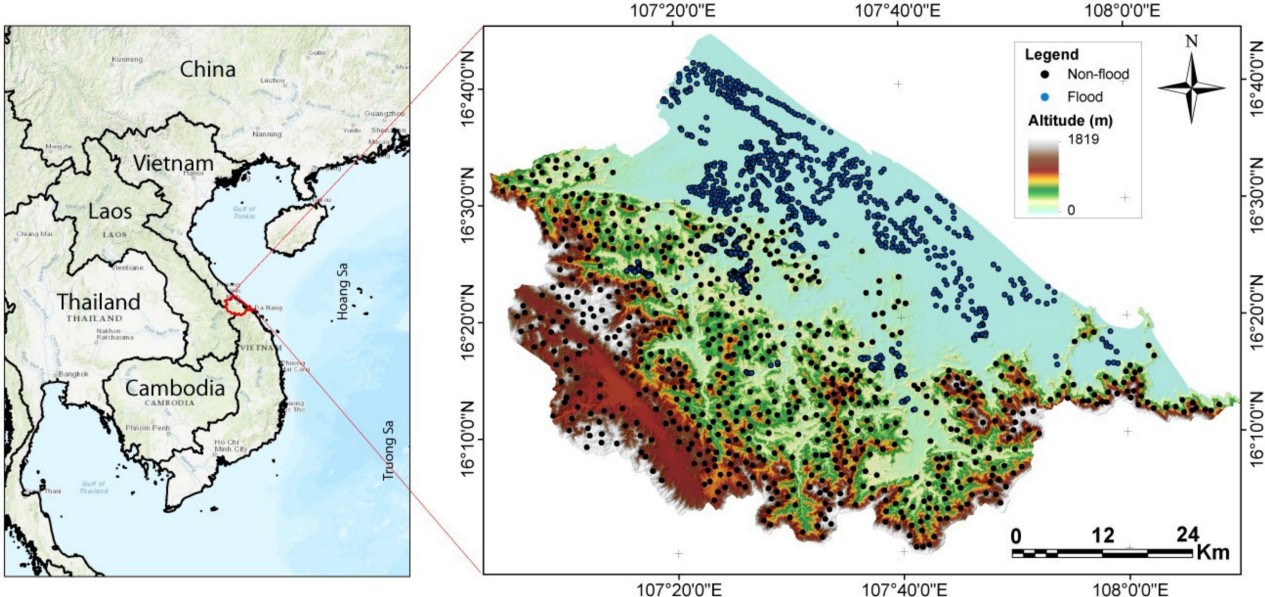

**Figure 1.** Study area location.

The region has a tropical monsoon climate with two main seasons: the dry season lasts from March to August, the rainy season from September to February. As a result, Thua Thien Hue has some of the highest rainfall in the country. Average annual precipitation can vary from 2500 to 3500 mm in the plains and 3000 to 4500 mm in the mountains, with 70% of the yearly rainfall concentrated during the rainy season.

70% of the area of Thua Thien Hue province is covered by forest. However, in recent years, these areas have been greatly reduced due to the development of agriculture, forest fires and illegal deforestation. According to the data from the Department of Natural Resource and Environment in Thua Thien Hue, approximately 800 ha of the forest has been diminished from 2018 to 2019. This is considered to be one of the important causes of flooding. In October 2020, several tropical cyclones, including Linfa, Nangka, Ofel, Saudel, and Molave, brought strong winds and heavy rainfall that caused historic floods and severe landslides in Thua Thien Hue and other provinces in central Vietnam. According to the Center for Disaster Prevention and Search and Rescue in Vietnam, on 9 October the mountainous districts of the province received the heaviest rainfall in the region, especially the Bo River in Phu Oc, which exceeded 5.4 m, higher than the previous peak of 0.22 m in 1999. On 18–19 October 2020, Thua Thien Hue saw particularly heavy rain, with rainfall of 150–300 mm and heavy flooding which killed more than 30 people and submerged hundreds of thousands of houses, causing an estimated 86.29 million USD in damage.

## 3. Methodology

### 3.1. Geospatial Database

#### 3.1.1. Flood Inventory Map

Flood inventory maps play an essential role in the development of a flood susceptibility model. They present information on the locations of past flooded areas and can predict future floods by analyzing relationships between past floods and their conditioning variables [53,54]. Various data sources were used to ascertain past flood locations, including official government sources and flood marks collected by the field mission.

In addition, our study used the remote sensing technique to extract flood locations. Two Sentinel 1A images before the flood (20 September 2020) and after the flood (18 October 2020) were used to detect flooded areas, by comparing the two images. This process is divided into four main stages [55,56]:

(i)　GRD Sentinel-1 products had not received radiometric pixel corrections and radiometric bias may still have been present in the image. Therefore, this image had to be calibrated to convert the pixel values of the digital values recorded by the sensor into a backscattering coefficient in order to be able to compare the images acquired on different dates.

(ii)　Speckle filtering to increase the readability of the image was an important step. A filtering operation consisted of estimating the ideal radar reflectivity area as a function of the noisy observation and taking into account the statistical parameters of the locally estimated scene. Many filters such as Lee, Gramma Map, the Nathan, Lee-Sigma18, Frost, and Refined Lee were used in previous studies. However, in this study, the Lee filter was used to suppress noise because it reduces the quality of the SAR image.

(iii)　After the pre-treatment process, flooded areas were determined using binarization to create a new binary image of water and non-water.

(iv)　529 flood points were obtained in the flood zone. In addition, 529 non-flood points were randomly selected from the non-flood zone in order to reduce bias.

#### 3.1.2. Flood Conditioning Factors

The selection of conditioning factors is considered an important task; scientifically selected factors will greatly improve the accuracy of a flood susceptibility model. These factors are divided into environmental, hydrological, climatic, and human activities [25,57]. In this study, 14 conditioning factors were selected, namely elevation, aspect, curvature, slope, density of river, density of road, distance to river, distance to road, flow direction,

Topographic Wetness Index (TWI), Normalized Difference Vegetation Index (NDVI), Normalized Difference Built-up Index (NDBI), Normalized Difference Water Index (NDWI), and rainfall (Figure 2).

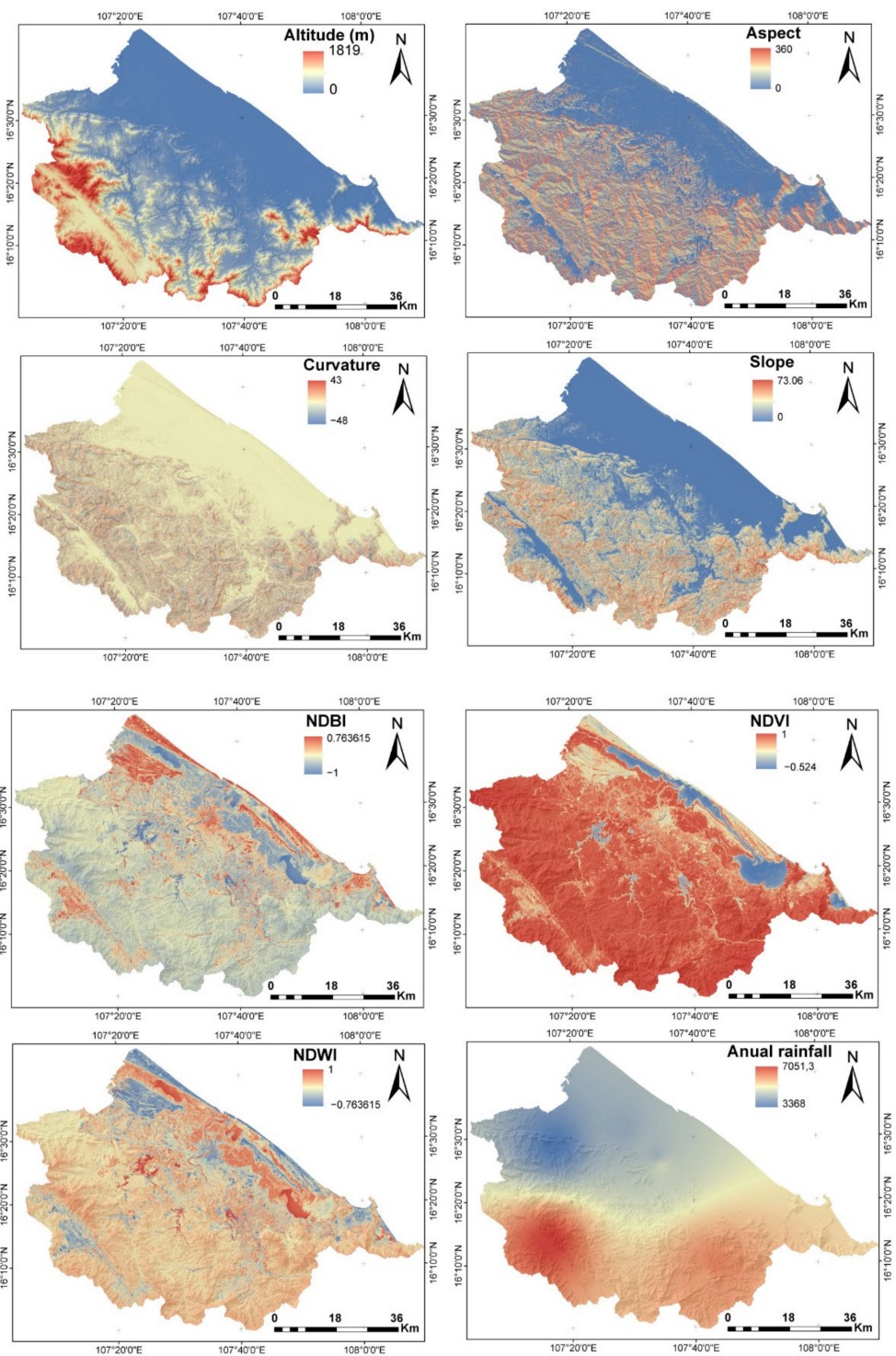

**Figure 2.** *Cont.*

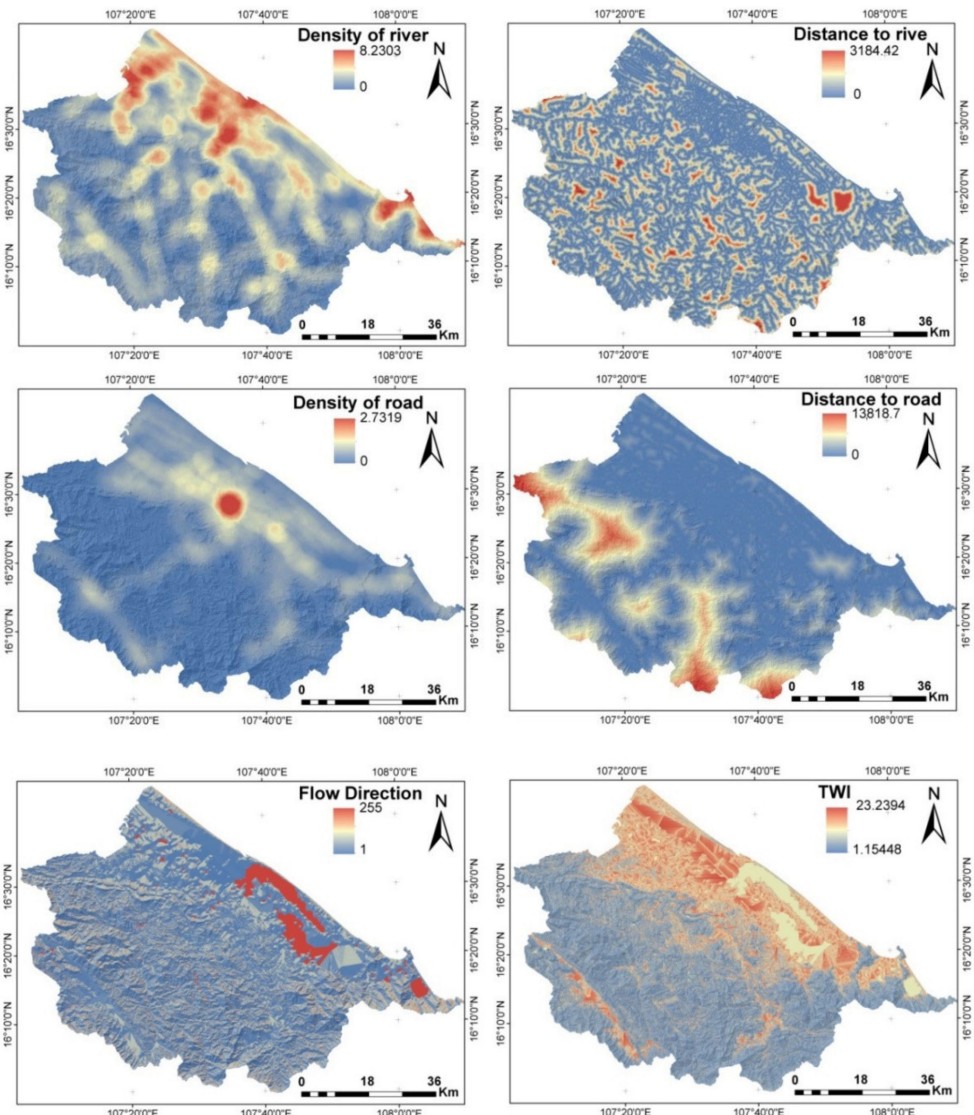

**Figure 2.** Variable prediction in study area.

Elevation, aspect, curvature, slope, TWI, and flow direction were computed from the Digital Elevation Model (DEM), which was built from a topographic map with a scale of 1/50,000, available from the Department of Natural Resources and the Environment of Thua Thien Hue province. Density of river, density of road, distance to river, and distance to road were computed from a topographic map with a scale of 1/50,000. The annual rainfall in 2020 was constructed based on ten climatic and hydrological stations in the province. These data were used to construct the precipitation map by applying kriging methods.

Elevation is considered to be one of the most important factors in flood susceptibility modeling because it directly influences the condition of the flow. The value of the elevation is the proportionality of the occurrence of the flood [58], while slope has an important relationship with both water accumulation and flow rate. The water accumulation capacity and the flow speed increase with the slope value [59,60].

Aspect is an essential factor in flood susceptibility analysis because it presents the direction of slope and flow [61], whereas runoff concentration is defined by curvature. The probability of flooding occurring is often concentrated on lower curvatures [62,63].

TWI is a crucial hydrological factor, which influences the process of flow accumulation [57,64]. The following equation computes TWI:

$$TWI = \ln(\alpha/\tan\beta) \tag{1}$$

where $\alpha$ is by upslope and $\beta$ is by slope.

Distance to river and density of river are the critical factors influencing the probability of flood occurrence because they control the propagation and intensity of the flooding. If the distance to the river is short, the probability of flooding will be high. Regions where the density of the river is higher are more susceptible to flooding [1,5,60].

Flow direction is one of the most important parameters to consider when establishing hydrological modeling and flood prediction. It describes the direction of the water and its distribution on the surface [65,66].

Rainfall is considered a trigger for flooding, as highlighted in several studies [60]. This is because rain causes overflows of river holding capacity and accumulated sediment, one of the major causes of flooding [1,67].

NDVI reflects vegetation density. Propagation and intensity of flooding depend on vegetation density [68]. NDWI is also used to represent changes related to the water content on the earth's surface [25]. The values of NDVI and NDWI are computed from Landsat 8 OLI in 2020 with the following equations:

$$NDVI = (NIR - RED)/(NIR + RED) \tag{2}$$

$$NDMI = (NIR - SWIR)/(NIR + SWIR) \tag{3}$$

where NIR, RED, and SWIR are the surface reflectance of the near infrared, red, and short-wavelength infrared.

Human activities such as road construction (and hence changes in distance to road and density of road) and house construction (as it relates to NDBI) can significantly influence the probability of flooding, especially as a result of increase in flow rate and decrease in water surface permeability [69,70].

*3.2. Machine Learning Methods*

The methodology used to construct a flood susceptibility map in this study is divided into four main steps: (i) data preparation, (ii) construction of model, (iii) validation of model, and (iv) flood susceptibility map analysis (Figure 3).

In the first step, the inventory map (flood and non-flood points) and 14 conditioning factors (DEM, aspect, curvature, slope, TWI, distance to river, distance to road, density of river, density of road, flow direction, NDVI, NDBI, NDWI and rainfall) were constructed from different sources. Among them, the flood points were roped by 1, while the non-flood points were roped by 0. Next, Random Forest was used to prioritize the influencing factors to reduce data redundancy and focus on the essential factors. Finally, this data is used as the input data into the flood susceptibility model.

In the second step, the data was divided into three separate groups: 60% of the data was randomly selected to train the models, another set with 20% of the data to validate the models, and 20% to test the model to ensure that it could be applied in practice. Then the BES was used to optimize the parameters of four individual models: SVM, BA, RF, and MLP.

In the third step, model validations represent the evaluation of performance that trains with specified data. In this study, various statistical indices were used to validate the model, such as AUC, RMSE, Accuracy, and MAE. In addition, the performance of the proposed models was compared with the reference models to ensure the utility and the alternative capacity of the model. A separate dataset with 324 flood and non-flood points was used to validate the models.

After validation, eight models were used to construct the flood susceptibility map in Thua Thien Hue province. This process was achieved by providing the models of the entire study area with 14 prediction variables. The output values vary from 0 to 1 and are classified into five groups: very low, low, moderate, high, and very high, using the natural break method.

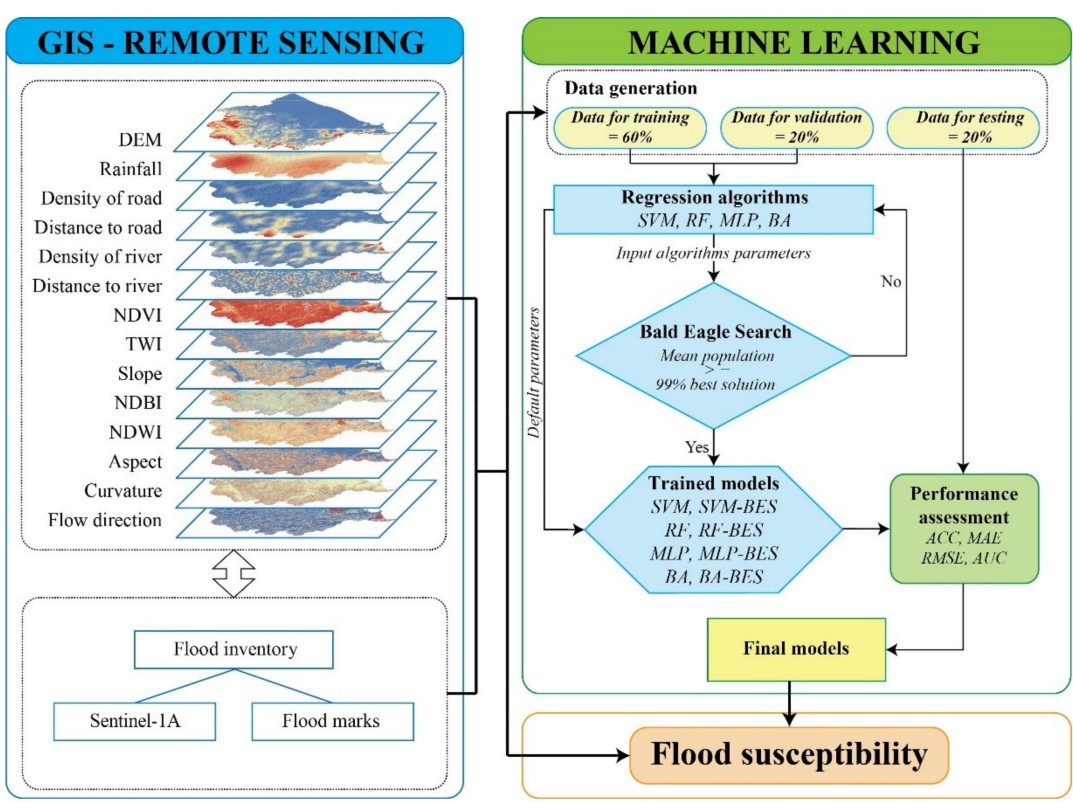

**Figure 3.** Flow chart of the flood susceptibility mapping process.

### 3.2.1. Support Vector Machine (SVM)

SVM is a family of machine learning algorithms that solve classification, regression, and anomaly detection problems and were developed in the 1990s by Vapnik [71]. They aim to separate data into classes in order to use a border that is as "simple" as possible. The distance between the different groups of data and the border between them is maximum. This distance is thus qualified as "wide margin separators", with the "support vectors" being the data closest to the border. SVMs often rely on the use of "kernels". These mathematical functions separate the data by projecting them in feature space (a vector space of greater dimensions). The margin maximization technique makes it possible to guarantee better robustness against noise and, therefore, a more generalizable model

Regression by Support Vector Machines (SVR) consists in finding the function f(x), which has at most one deviation $\varepsilon$ from the learning examples (xi, yi), for i = 1, . . . , N and which is as flat as possible. This amounts to not considering errors smaller than $\varepsilon$ and prohibiting those greater than $\varepsilon$ [72]. Maximizing the flatness of the function makes it possible to minimize the complexity of the model, which influences its performance in generalization. Thus, SVM methods are based on controlling the complexity of the model during training. During SVM model development, two parameters (gramma and kernel) must be optimized.

### 3.2.2. Random Forest (RF)

RF is a supervised learning algorithm that can be used for both classification and regression, as proposed by [73]. This algorithm is based on assembling decision trees and working independently on a vision of a problem. Each tree is an estimator which has a fragmented view of the problem. All of the trees are brought together to give an overall estimate. It is the assembly of different opinions which makes the prediction extremely powerful. Each model is randomly distributed to the subsets of decision trees. This algorithm works in four main steps: (1) randomly select the sample from the dataset; (2) construction of decision trees for each sample; (3) vote for the predictions; (4) determi-

nation of the decision trees with the most votes. In this study, the input data into the RF model was strung through 500 decision trees to reach the final model.

### 3.2.3. Bagging (BA)

BR is an artificial intelligence technique introduced by [74] that can be used for both classification and regression problems, which consists of assembling a large number of "weak learners" to create a "strong learner". This is done by voting. That is, each weak learner will cast one vote and the strong learner prediction will be the average of all responses given [75,76]. This algorithm can improve the ability to generalize and decrease the error in the classification process.

### 3.2.4. Multilayer Perceptron (MLP)

MLPR is a directed network of artificial neurons organized in several layers, where information flows from the input layer to the output layer only. Multilayer Perceptron is organized in a hierarchy: an input layer includes several neurons which are equivalent to the explanatory variables. This output layer has the most neurons and one or more intermediate hidden layer levels [77–79]. The input layer always represents a virtual layer associated with system inputs. It does not contain any neurons. The next layers are layers of neurons. Each layer consists of a variable number of neurons; the neurons of the output layer always correspond to the system's outputs. The connections between neurons are never made on the same layer, but from one layer to another, by weighted connections. The weights are generated by the operation of the network and connect the space of the inputs to the space of the outputs using a nonlinear transformation [80]. The following equation can describe each neuron output:

$$Yi = f\left(\sum Wij\ Xi\right) \tag{4}$$

where $Yi$ are the input values by the node $j$. The function $f$ can be the function threshold, sigmoid, or hyperbolic tangent. The weights between node $i$ and $j$ are $wij$ and $xi$ are the output results at node $i$.

### 3.2.5. Bald Eagle Search Optimization Algorithm (BES)

BES is a novel meta-heuristic optimization, inspired by nature. This algorithm describes bald eagles' hunting strategy when looking for fish [49], such as salmon. Their hunting strategies consist of three distinct processes: the first is selecting the region within which to search for prey, the second is the search, and the third is the attack [49].

The bald eagle identifies and selects the best region in its search space to hunt for prey during the selection process. The previous movements identify each sound movement.

$$P_{new,i} = P_{best} + \alpha \times r(P_{mean} - P_i) \tag{5}$$

where $P$ is the location of the bird, $\alpha$ is a parameter to control the change of location of the bird—which ranges from 1.5 to 2—and $r$ is the random value—with a range of 0 to 1. $P_{best}$ is considered the best location of the eagle in the past and $P_{mean}$ is all of the previous research space.

Then, from the region selection process, the eagle searches for prey in their search spaces in a spiral pattern with different sizes.

$$P_{i,\ new} = P_i + y(i) \times (P_i - P_{i+1}) + x(i) \times (P_i - P_{mean}) \tag{6}$$

$$x(i) = \frac{xs(i)}{\max(|xs|)}$$

$$y(i) = \frac{ys(i)}{\max(|ys|)}$$

$$xs(i) = s(i) \times \sin \theta_i$$

$$ys(i) = s(i) \times \sin \theta_i$$

$$\theta_i = \beta \times \prod \times rd$$

$$s(i) = \theta_i + R \times rd$$

where $\beta$ is the position of the corner, ranging from 5 to 10. Thus, $r$ is the search cycle—ranging from 0.5 to 2—and $rd$ is the random value.

After the second process, the bald eagle attacks the prey. The following equations describe this process:

$$P_{i,\ new} = rd \times E_{best} + xt(i) \times (E_i - b1 \times E_{mean}) + yt\ (i) \times (E_i - b2 \times E_{best}) \tag{7}$$

$$xt(i) = \frac{xs(i)}{\max(|xs|)}$$

$$yt(i) = \frac{ys(i)}{\max(|ys|)}$$

$$xs(i) = s(i) \times \sin \theta_i$$

$$ys(i) = s(i) \times \sin \theta_i$$

$$\theta_i = \beta \times \prod \times rd$$

$$s(i) = \theta_i$$

where $b1$ and $b2$ range from 1 to 2.

### 3.3. Accuracy Assessment

In this study, eight flood susceptibility models were constructed using the same training and validation data. Then, their performance was evaluated and compared by applying various statistical indices, such as Root Mean Square Error (RMSE), Receiver Operation Characteristics (ROC), Mean Absolute Error (MAE), and Accuracy.

RMSE is used to measure the differences between values (sample or population values) predicted by a model or estimator, and observed values (or true values). The RMSE is always positive and a value of 0 (almost never reached in practice) would indicate a perfect fit to the data. In general, a smaller RMSE value indicates better accuracy than a higher RMSE value [81].

$$\text{RMSE} = \sqrt{\frac{1}{n} \Sigma_{i-1}^n \left( X_{predicted} - Y_{actual} \right)^2} \tag{8}$$

The mean absolute error (MAE) is the arithmetic mean of the absolute values of the deviations. It represents the difference between the values fitted by the model and the observed data [82].

$$\text{MAE} = \frac{1}{n} \Sigma_{i-1}^n \left| X_{predicted} - Y_{actual} \right| \tag{9}$$

where $n$ is the number of samples in the database, $X_{predicted}$ is the predicted value of the dependent variable for sample $i$, and $Y_{actual}$ is the observed, expected value for item $i$.

The ROC curve is a graphical representation of the relationship between the sensitivity and 1-specificity of a test for all possible threshold values [83,84]. The region under the curve (AUC) is used to assess the accuracy of the model, ranging from 0 to 1. The higher the AUC value, the higher the performance of the model.

$$\text{AUC} = \sum \text{TP} + \sum \frac{\text{TN}}{\text{P}} + \text{N} \tag{10}$$

Accuracy is measured as the rate of correctly classified individuals out of the total number of individuals [85].

$$ACC = \frac{TP + TN}{TP + TN + FP + FN} \qquad (11)$$

where TP and FP are the number of pixels classified correctly and incorrectly as flood, respectively, and TN and FN are the correctly classified and incorrectly non-flooded pixel, respectively. P, N are the number of flood and non-flood samples.

## 4. Results

### 4.1. Spatial Relationship

The Random Forest model was used to hierarchize the 14 conditioning factors considered in this study (Figure 4). Based on Random Forest, the VI index was calculated for each conditioning factor and these factors were ranked from 1 to 14. The results show that all factors contributed to the flood prediction. Among them, DEM (0.0699), rainfall (0.0146), density of river (0.0118), and distance to road (0.0117) are more influential on the spatial modeling of flood susceptibility compared to other factors. Then, density of road (0.0105), distance to river (0.0094), NDVI (0.0086), TWI (0.0084), slope (0.0071), and NDBI (0.0071) are classified from fifth to tenth. The other four factors: NDWI (0.0062), aspect (0.0043), curvature (0.0013), and flow direction (0.001) have the least effect on the flood susceptibility model.

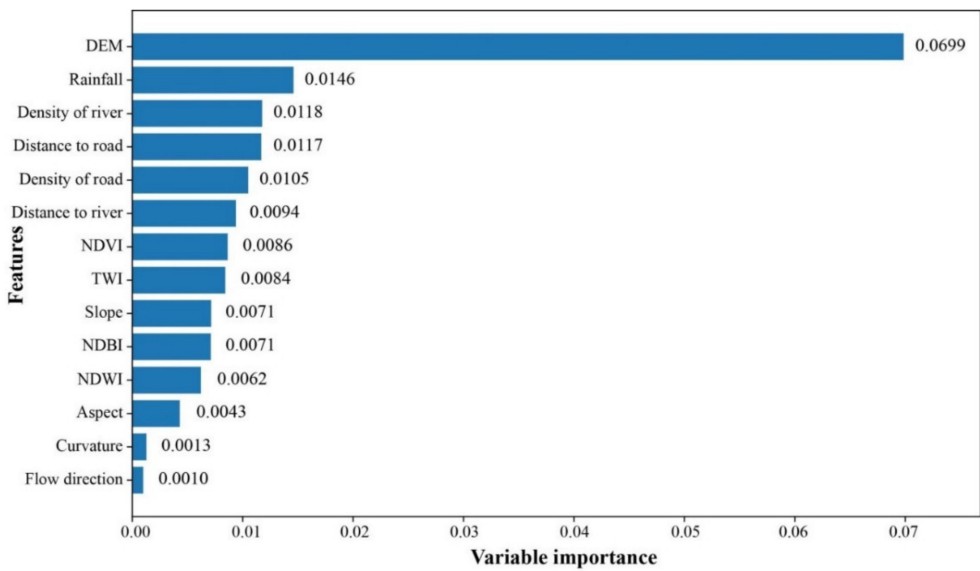

**Figure 4.** Variable importance using Random Forest.

Another important step is building the model with optimization parameters to ensure the ability to re-experiment in other regions around the world. Table 1 represents the parameters of the four models (SVM, RF, BA, and MLP) which are optimized by the BES algorithm. Searches for optimal values were assigned using a trial and error method.

### 4.2. Model Performance Comparison

The flood susceptibility models were trained with the same training data and evaluated using the same validating data and testing data. Figure 5 shows the ROC curve of eight prediction models, namely SVM, RF, BA, MLP, SVL-BES, RF-BES, BA-BES, and MLP-BES. Among them, the RF-BES and BA-BES modes were more efficient with AUC = 0.998. The MLP-BES model was second with AUC = 0.995. The SVM-BES model was third with AUC = 0.994. The MLP, and SVM models were fourth with AUC = 0.993. The remaining two models, RF and BA, performed less well than the other models, with AUC = 0.992 and

0.986, respectively. The three models RF-BES, BA-BES, and RF—with AUC = 0.999 were most efficient for testing data. Next were the MLP-BES with AUC = 0.998, BA and MLP with AUC = 0.997, and SVM with AUC = 0.995. Finally, the SVM-BES model was least efficient, with AUC = 0.991.

**Table 1.** Optimization parameter of models using BES.

| Algorithm | Parameter | Value Ranges | Best Value | Mean Value |
|-----------|-----------|--------------|------------|------------|
| SVM | C | 0.1–100 | $8.7146 \times 10^1$ | $8.7011 \times 10^1$ |
| | gamma | 0.0001–10 | $4.2251 \times 10^{-1}$ | $4.2269 \times 10^{-1}$ |
| RF | max_features | 1–14 | $1.5911 \times 10^0$ | $1.8374 \times 10^0$ |
| | n_estimators | 1–1000 | $4.0651 \times 10^1$ | $4.1091 \times 10^1$ |
| | min_samples_split | 2–100 | $3.2807 \times 10^0$ | $3.6000 \times 10^0$ |
| | min_samples_leaf | 1–100 | $1.4900 \times 10^0$ | $1.2565 \times 10^0$ |
| BA | max_features | 1–14 | $8.0361 \times 10^0$ | $7.0130 \times 10^0$ |
| | n_estimators | 1–1000 | $9.9618 \times 10^0$ | $1.7403 \times 10^2$ |
| MLP | hidden_layer_sizes | 1–200 | $1.4563 \times 10^2$ | $1.2727 \times 10^2$ |
| | alpha | 0.0001–1 | $1.0341 \times 10^{-2}$ | $7.1631 \times 10^{-3}$ |
| | max_iter | 100–1000 | $8.4141 \times 10^2$ | $7.7514 \times 10^2$ |

In addition to the ROC, various statistics, including Accuracy, RMSE, and MAE were used to evaluate the performance of the models (Table 2). For the validation phase, in general, four hybrid models (MLP-BES, BA-BES, RF-BES, and SVM-BES) scored higher than the individual models. Among them, BA-BES was best, with Acc = 0.91, RMSE = 0.018, and MAE = 0.053, followed by RF-BES (Acc = 0.9, RMSE = 0.019, and MAE = 0.06), MLP-BES (Acc = 0.88, RMSE = 0.025, and MAE = 0.07), SVM-BES (Acc = 0.86, RMSE = 0.029, and MAE = 0.09), respectively.

**Table 2.** Model performance and comparison.

| Methods | Validating Data | | | | Testing Data | | | |
|---------|----------|------|-----|-----|----------|------|-----|-----|
| | Accuracy | RMSE | MAE | AUC | Accuracy | RMSE | MAE | AUC |
| SVM | 0.8483 | 0.0334 | 0.1070 | 0.993 | 0.8225 | 0.0310 | 0.1179 | 0.9954 |
| RF | 0.8676 | 0.0291 | 0.0496 | 0.992 | 0.9330 | 0.0117 | 0.0327 | 0.9989 |
| BA | 0.8637 | 0.0300 | 0.0481 | 0.986 | 0.9178 | 0.0144 | 0.0349 | 0.9968 |
| MLP | 0.8623 | 0.0303 | 0.0937 | 0.993 | 0.8442 | 0.0272 | 0.0988 | 0.9971 |
| SVM-BES | 0.8645 | 0.0298 | 0.0993 | 0.994 | 0.7738 | 0.0395 | 0.1337 | 0.9908 |
| RF-BES | 0.9095 | 0.0199 | 0.0611 | 0.998 | 0.8669 | 0.0232 | 0.0737 | 0.9986 |
| BA-BES | 0.9159 | 0.0185 | 0.0537 | 0.998 | 0.9174 | 0.0144 | 0.0503 | 0.9992 |
| MLP-BES | 0.8861 | 0.0251 | 0.0767 | 0.995 | 0.8713 | 0.0225 | 0.0857 | 0.9985 |

For the individual models, RF was most efficient, with Acc = 0.86, RMSE = 0.029, and MAE = 0.04, followed by BA (Acc = 0.86, RMSE = 0.03, and MAE = 0.04), MLP (Acc = 0.86, RMSE = 0.03, and MAE = 0.09) and SVM (Acc = 0.84, RMSE = 0.033, and MAE = 0.1), respectively.

For the testing phase, the RF model was more efficient than the other models with Acc = 0.93, RMSE = 0.01, and MAE = 0.03, followed by BA (Acc = 0.91, RMSE = 0.01, and MAE = 0.03), BA-BES (Acc = 0.91, RMSE = 0.01, and MAE = 0.05), MLP-BES (Acc = 0.87, RMSE = 0.02, and MAE = 0.08), RF-BES (Acc = 0.86, RMSE = 0.02, and MAE = 0.07), and MLP (Acc = 0.84, RMSE = 0.02, and MAE = 0.09), SVM (Acc = 0.82, RMSE = 0.03, and MAE = 0.11) and SVM-BES (Acc = 0.77, RMSE = 0.03, and MAE = 0.13), respectively.

The results show that all the proposed models performed with AUC > 0.9 and Accuracy >77% with the validation and testing datasets, demonstrating that these models represent good performance and a high capacity for generalization.

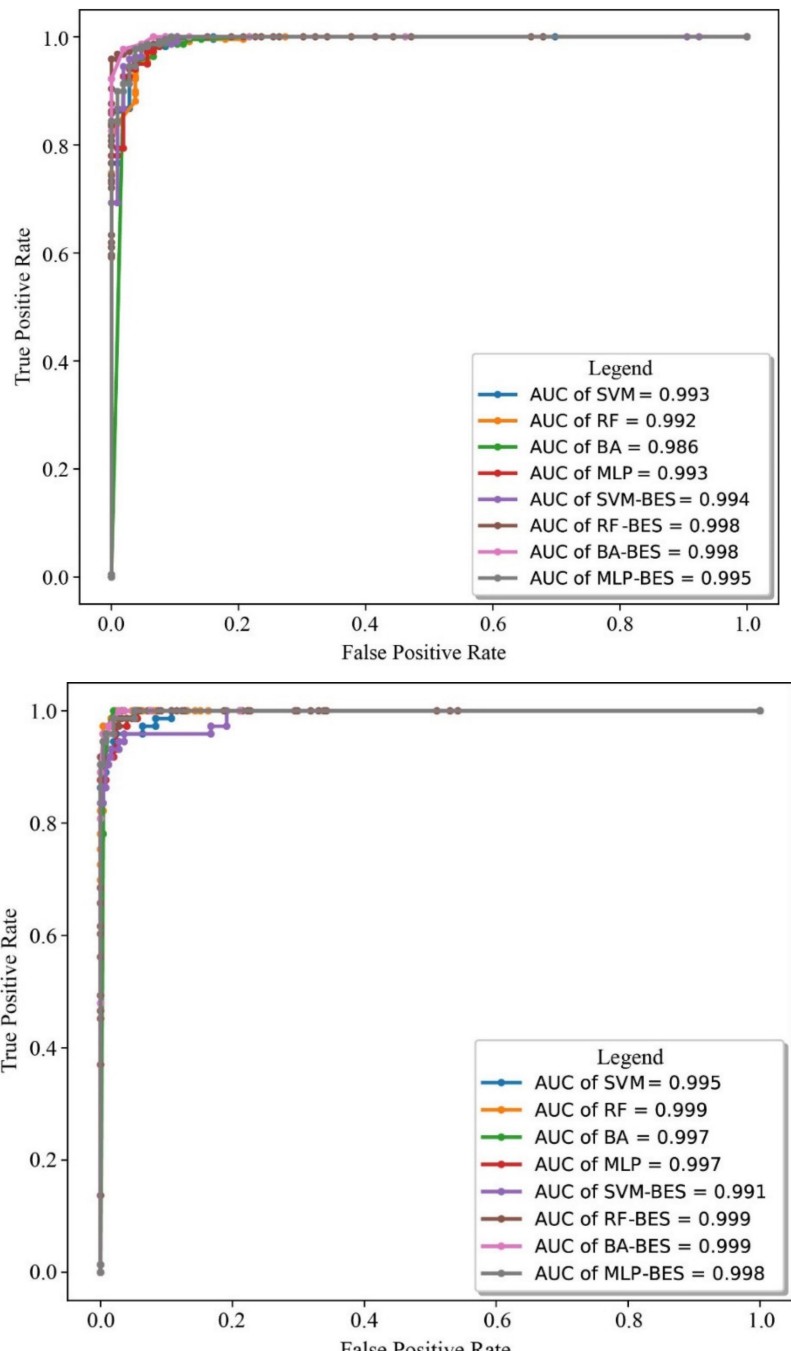

**Figure 5.** ROC in the model validating (top) and testing (bottom).

*4.3. Flood Susceptibility Map*

Figure 6 shows the susceptibility map constructed by the model SVM, RF, BA, MLP, SVM-BES, RF-BES, BA-BES, and MLP-BES. The distribution of each class with the different models is presented in Table 3. Although there are performance differences between models, all models showed that most high and very high flooding occurs on the coastal plain and in low areas along the river. Among them, for the SVM model, 28.57% of the study area was located in areas with high and very high flood susceptibility; this figure was 39.13% for the RF model, 41.93% for BA, 33.42% for MLP, 32.84% for SVM -BES, 31.55% for RF-BES, 39.41% for BA-BES, and 33.17% for MLP-BES.

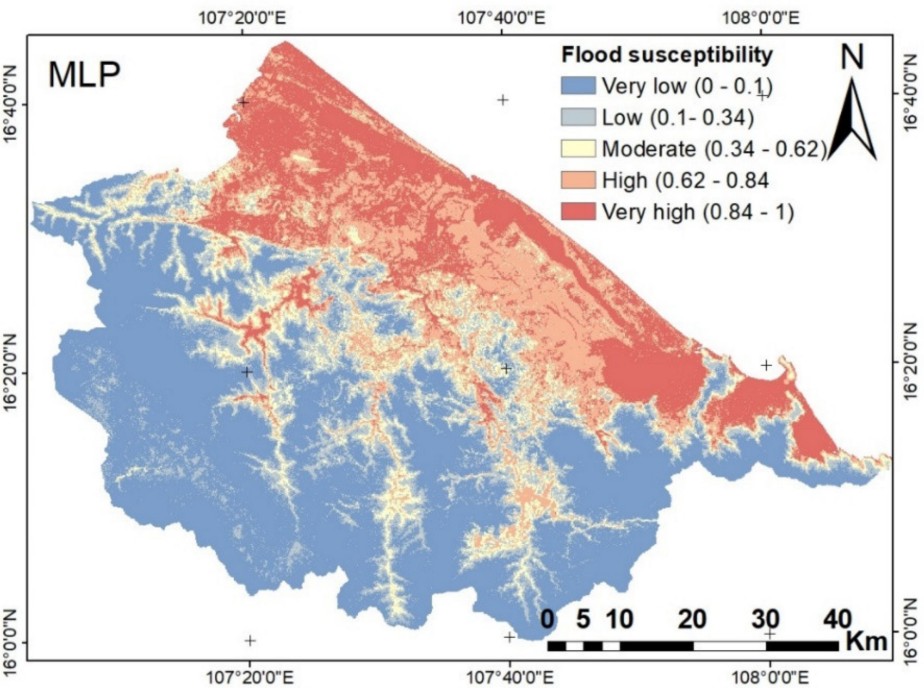

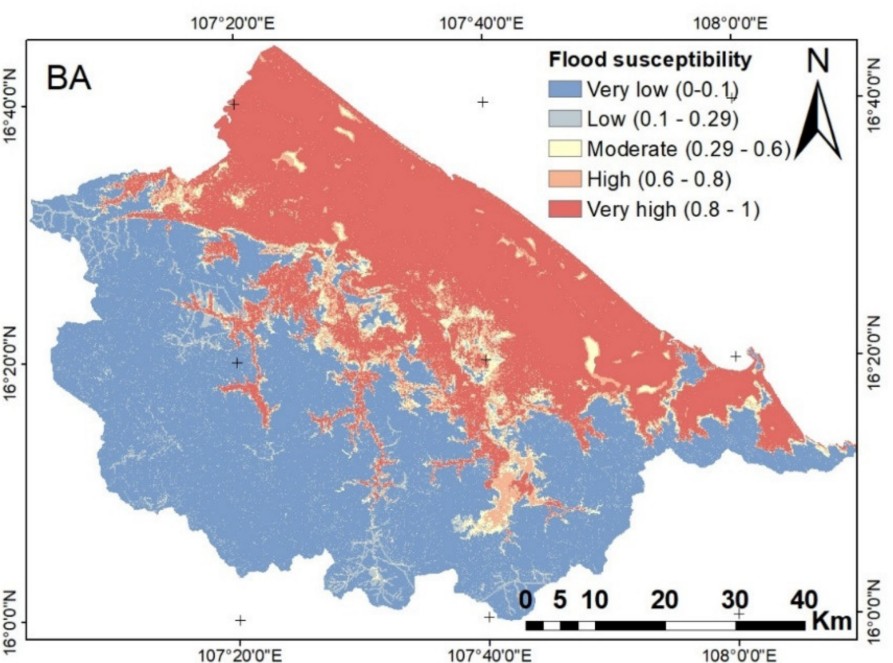

**Figure 6.** *Cont.*

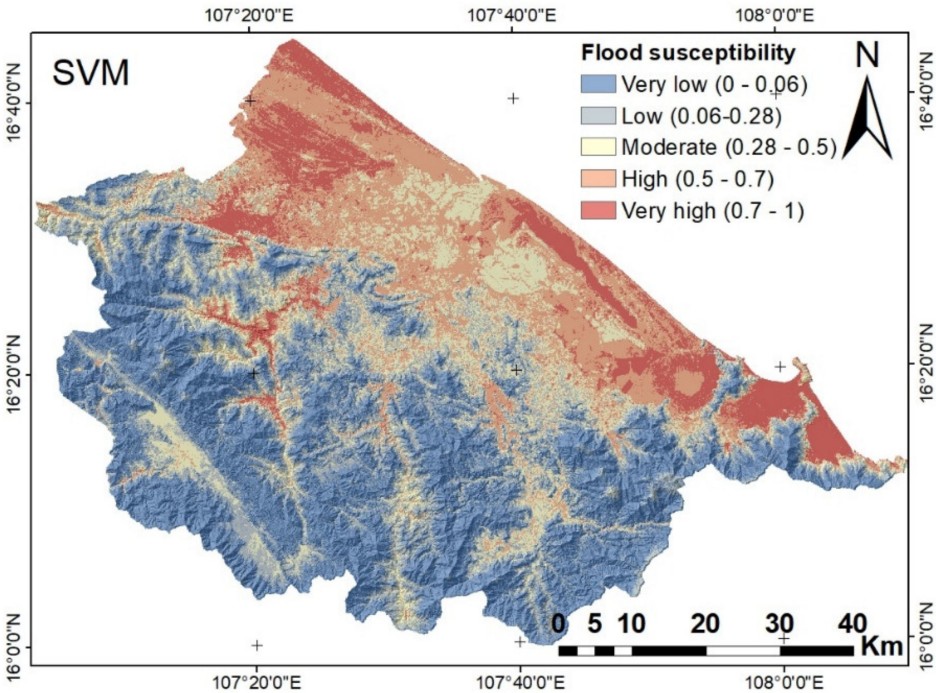

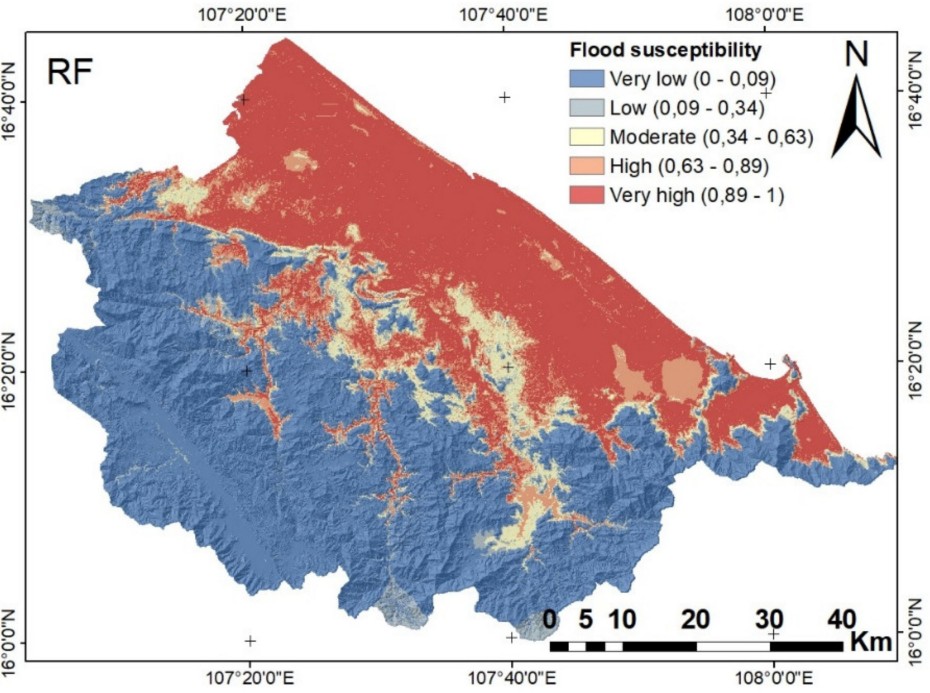

**Figure 6.** *Cont.*

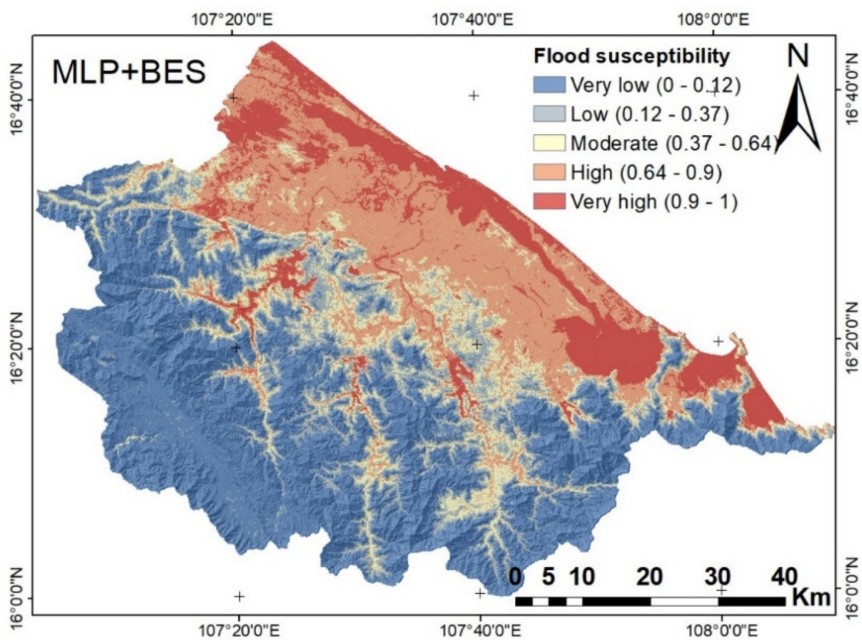

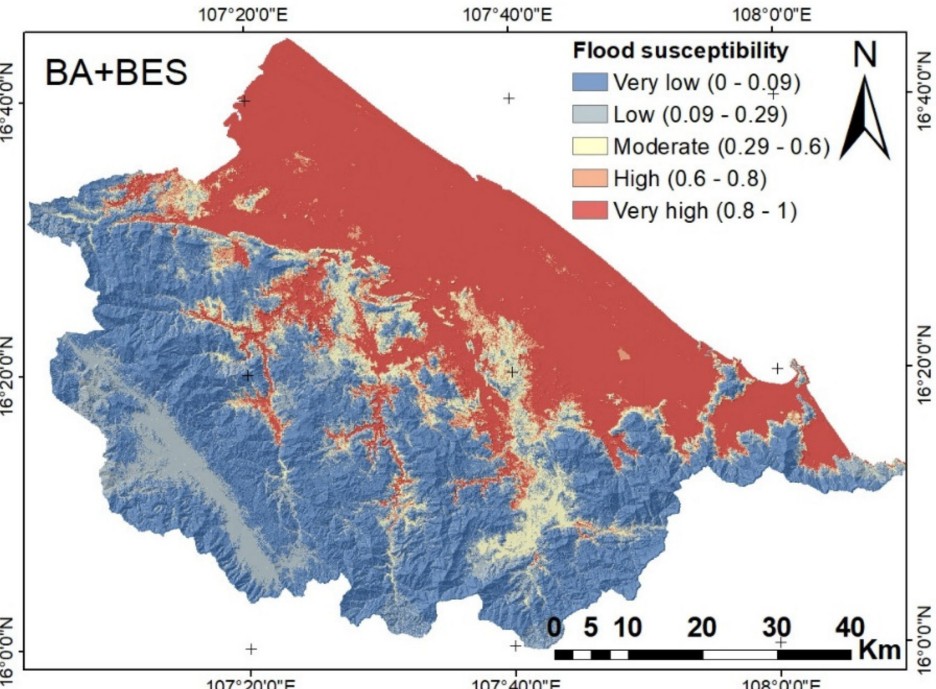

**Figure 6.** *Cont.*

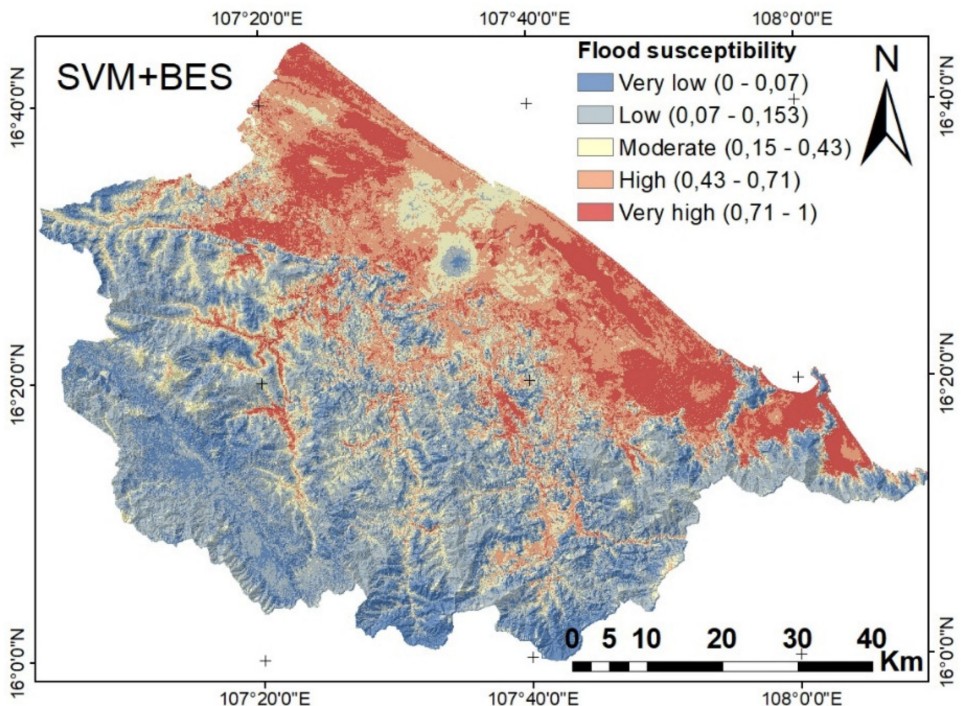

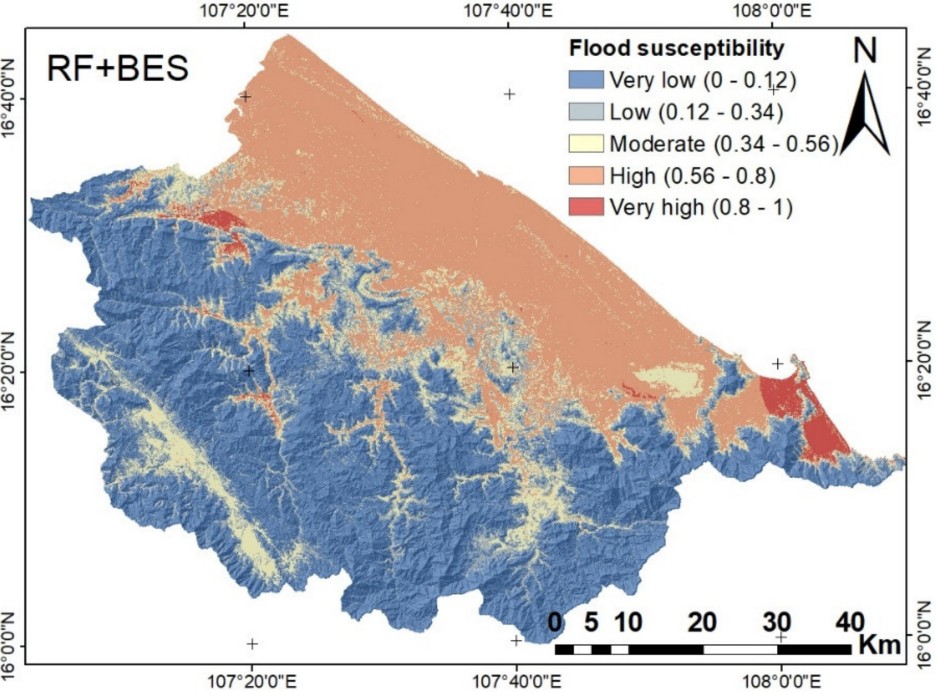

**Figure 6.** Flood susceptibility in the Hue province.

**Table 3.** The distribution of each class with the different models.

| Methods | Very Low (km$^2$) | Low (km$^2$) | Moderate (km$^2$) | High (km$^2$) | Very High (km$^2$) |
|---|---|---|---|---|---|
| SVM | 1742.601 | 979.457 | 782.9179 | 915.818 | 486.3461 |
| RF | 2461.101 | 231.1001 | 294.6731 | 432.7302 | 1487.536 |
| BA | 2357.269 | 257.8551 | 234.6635 | 278.8325 | 1778.5 |
| MLP | 2023.755 | 724.264 | 518.9837 | 783.2094 | 856.9273 |
| SVM-BES | 854.2445 | 1634.21 | 807.2763 | 919.9767 | 691.4326 |
| RF-BES | 2265.415 | 571.288 | 522.1422 | 1472.856 | 75.4391 |
| BA-BES | 1917.047 | 722.5044 | 333.5081 | 196.8266 | 1737.253 |
| MLP-BES | 2099.294 | 643.6636 | 536.4444 | 1023.34 | 604.3974 |

## 5. Discussion

Instances of flooding have been increasing both in intensity and frequency in the context of climate change. Therefore, assessing the probability of flood occurrence plays an ever more important role in establishing effective flood management strategies [53,86]. In this regard, with the rapid advancements in computer technology and remote sensing techniques, there are more and more opportunities to predict floods. The objective of this research is the development of a new approach based on state-of-the-art machine learning and remote sensing, namely BES, SVM, RF, BA, and MLP, to analyze flood susceptibility in the central Vietnamese province of Thua Thien Hue. The exact identification of these areas is essential, as authorities will use this to assess flood risk and formulate appropriate measures to minimize future damage.

This study generated an inventory map using radar image, due to the "all-weather" acquisition capacity of this image. The extraction of water from radar images is relatively easy and can be achieved by a simple radiometric thresholding technique [87,88]. However, the radar image also has disadvantages related to the difficulty of determining radiometric thresholds in the presence of strong wind at the date of acquisition: waves formed on the surface of the water tend to increase backscattering [89]. The values of 14 conditioning factors were used as the input data for the model. It should be mentioned that the correct selection of conditioning factors plays an important role in improving the model's performance because data redundancy can make the model more complex and affect the model's predictability. Therefore, Random Forest was used to assess the importance of these factors. The results show that all factors contributed to the prediction of flooding likelihood. Among them, the most influencing factors were DEM (0.0699), rainfall (0.0146), density of river (0.0118), and distance to road (0.0117). This result is entirely in line with the reality of Thua Thien Hue province in general and Vietnam in particular [10,90], as floods often occur in low areas with high river density. In addition, the extent of flooding depends on the height of the water, the speed of the current and the duration of the flood. These parameters are conditioned by precipitation. However, this result also contradicts some previous studies. [91] assessed the significance of rainfall to be fifth after elevation, distance to river, land use, and slope in the model of flood susceptibility in the Haraz watershed in Mazandaran Province of northern Iran. [92] considered rainfall to be ninth out of ten factors used. [93] highlighted that rainfall was ranked 10th out of 12 factors used to predict flood susceptibility in the Buzău river catchment of Romania. [94] reported that distance to river, slope, elevation, lithology, and soil were the most critical factors; rainfall was ranked sixth most important in the flood susceptibility model for the watershed of Tafresh county, Markazi province, Iran. There are two main reasons for this difference. The first, as highlighted by [5], is that the importance of the conditioning factors depends on natural characteristics and the methodology used. The second is related to the type of flood. There are two main types of flooding: The first type occurs as a result of high water in the river during the rainy season. It is an annual invasion of space by water, controlled by dikes and canals. Due to the low altitude and dense hydrological networks, submersions occur several times a year. In this case, although rainfall is the trigger for the

flood, it is not a determining factor because the flood occurs after a few days of rain. On the other hand, there is out-of-control flooding caused by extreme weather events, such as tropical storms [95]. In Thua Thien Hue province, flooding often occurs alongside tropical cyclones and depressions, causing heavy rains on a large scale. Therefore, precipitation and topography are the two most important factors.

Our result confirms that BES is successful in improving the performance of the base model of SVM, BA, RF, and MLP. As discussed above, with the complex and non-linear structure of flooding events, individual models easily lead to bias in flood susceptibility prediction results. Therefore, the nonlinear relationships between the conditioning factors and the flood point are minimized by adjusting the parameters, although the parameter adjustment can reduce these problems and improve the performance of the model. However, it should be noted that these tasks can be very difficult and time-consuming and can lead to the unfair comparison between models [49]. The tuning of the parameters itself is an optimization problem and is solved using a hyper parameter optimizer. In this study, we determined the range of values for each parameter of the individual algorithm to tune these parameters. The model runs 1000 iterations and their values increase with each iteration. Table 1 shows that most of the default values of the hyper parameters of the individual algorithms are always lower after applying the BES algorithm. The population mean value is quite asymptotic with respect to the best solution from BES. It shows that using the fine-tuning algorithm can significantly improve the performance of individual models. In general, among the four proposed models, BA-BES outperforms the other models, with AUC = 0.998 for validation and AUC = 0.9992 for testing. BA has the advantage of combining many weak learners into strong learners, and input data noises can be greatly reduced by decreasing the sensitivity of individual classifications with the Bootstrap sampling method. Another important advantage of BA is the modification of the generalization errors on the base classification [96]. BA can help to reduce the variance and so can eliminate the problem of overfitting. RF-BES also has high prediction capability because RF can automate missing values in the database and overfitting data [97]. However, the performance of RF-BES is slightly lower compared to BA-BES. This is not surprising because one of the important limitations of this algorithm is computation time and the difficulty in determining the significance of the predictor variables [98]. MLP-BES is third because this algorithm has too many parameters and each node is connected to other nodes, making the model more complex and inefficient [99]. Meanwhile, although the SVM-BES model has high predictability with AUC = 0.994, its performance is worse than the BA-BES, RF-BES, and MLP-BES models due to the difficulty of refining the models' hyper parameters C and Gamma [100].

In short, although there are minor differences in performance between the models proposed in this research, we have found that BES is effective in improving the performance of the basic classifications SVM, RF, BA, and MLP. From these results, we can conclude that a hybrid model approach can reduce the problems of noise, variance, and overfitting. Some point out that hybrid models take longer to build the final model compared to base models. However, the accuracy of flood prediction models should not be influenced by cost reduction strategies over time, which results in increased costs in the long run. Instead, strategies should aim to find new patterns to improve prediction accuracy, and support decision-makers when selecting the appropriate strategy for flood management, even in data-limited regions. This study does, however, demonstrate some of the general limitations characteristic to using machine learning to construct flood susceptibility maps. For example, these maps do not provide information on flood depth and flood velocity. In addition, although this study used DEM with a resolution of 10 m, in the future we will focus on the use of ASTER DEMs, as they better reflect the land area, particularly regarding the characteristics of buildings, vegetation, flow direction, and slope.

## 6. Conclusions

Flood susceptibility mapping with high accuracy is considered extremely important to establish strategies for flood management and sustainable land use planning, particularly in the face of climate change. The objective of this research is the development of a novel approach based on state-of-the-art machine learning and remote sensing—namely BES, SVM, RF, BA, and MLP to build flood susceptibility mapping in the Thua Thien Hue province. The findings are important from this perspective not only for the Vietnam, but for other countries often affected by flooding.

The integration of the BES algorithm with the individual model, namely SVM, BA, RF and MLP, can significantly improve the performance of the individual model. Among the proposed models, BA-BES was most effective with AUC = 0.998, followed by RF-BES (AUC = 0.998), MLP-BES (AUC = 0.998), SVM-BES (AUC = 0.99). The proposed new models with high accuracy can be used to construct the flood susceptibility map in any region, especially in data-limited regions.

Five levels of flood susceptibility, namely very low, low, moderate, high and very high have been divided in Thua Thien Hue province in Vietnam. Although there are small differences between the models, approximately 30–40% of the study area falls within the high and very high flood susceptibility zone. These main areas concentrate along the river and in the plain along the sea.

Flood risk management strategies are considered a top priority in the context of climate change and sea level rise, particularly in the tropical monsoon region where flooding often occurs. This study was applied in Thua Thien Hue province in Vietnam to determine the flood probability area using state-of-the-art methods such as machine learning and remote sensing. However, the finding of this study can be applied in other regions in the world.

**Author Contributions:** Conceptualization, M.C.H., G.Ş., I.R., P.B.; methodology, M.C.H., G.Ş., I.R., P.B.; software, T.P.H.; validation, M.C.H., P.L.V., T.B.H.D. and H.D.N.; formal analysis, T.P.H.; investigation, D.D.D.; resources, D.D.D.; data curation, T.P.H.; writing—original draft preparation, M.C.H., P.L.V., H.D.N., G.Ş., I.R., P.B.; writing—review and editing, M.C.H., H.D.N., G.Ş., I.R., P.B.; visualization, M.C.H., T.B.H.D., G.Ş., I.R., P.B.; supervision, M.C.H., P.L.V.; project administration, M.C.H.; funding acquisition, M.C.H. All authors have read and agreed to the published version of the manuscript.

**Funding:** This research was funded by VNU Science and Technology Development Fund (QG.21.33).

**Institutional Review Board Statement:** Not applicable.

**Informed Consent Statement:** Not applicable.

**Data Availability Statement:** Not applicable.

**Acknowledgments:** This research has been done under the research project QG.21.33 of Vietnam Nation University, Hanoi.

**Conflicts of Interest:** The authors declare no conflict of interest.

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
