# Peer review of "Machine Learning and Remote Sensing Application for Extreme Climate Evaluation: Example of Flood Susceptibility in the Hue Province, Central Vietnam Region"

_water, doi:10.3390/w14101617_

Round 1

Reviewer 1 Report

Main question addressed by the research: The work addresses the machine learning and remote sensing application for climate assessment in an example of flood susceptibility in the Hue province, Central Vietnam.
Originality and relevance of the topic: The topic is relevant to the field and it considers a suitable model (research gap).
Added value of the paper:  The manuscript takes into account the study of the machine learning and remote sensing application which is quite innovative, however the main purpose of it is not clearly stated and neither the direct applications.

Quality of figures: Very good

References: Up-to-date.

Improvements for the paper to be considered:

  1. I think they have covered all the aspects required for the results, however the validation of the results is not clearly explained and this is essential in order to justify the reliability of the results.
  2. Abstract is too short and general. It should summarize the main findings and applications of the paper.
  3. Lines 411-428 too descriptive and difficult to follow.
  4. The conclusions are a bit weak and they would need more elaboration so they clearly match the main findings. They have not highlighted the importance of the research.

Author Response

We would like to express our sincere gratitude to the reviewer for the positive and encouraging comments and for the overall appreciation of work. We have carefully read and addressed all comments, point by point, as indicated below.

Reviewer no. 1

Comment

Response

Reference

I think they have covered all the aspects required for the results, however the validation of the results is not clearly explained and this is essential in order to justify the reliability of the results.

We have corrected model validation as suggested.

Lines 252-269

Abstract

Abstract is too short and general. It should summarize the main findings and applications of the paper.

Thank you for your valuable question. We corrected the abstract as suggested part.

Lines 18-30

Lines 411-428 too descriptive and difficult to follow.

Corrected as suggested.

Lines 431-452

The conclusions are a bit weak and they would need more elaboration so they clearly match the main findings. They have not highlighted the importance of the research.

Thank you, we corrected the conclusion section as suggested.

Lines 582-607

Reviewer 2 Report

Dear Authors,

Thank you very much for the possibility to read your paper concerning Machine learning and Remote Sensing application for extreme climate evaluation. After reading that paper I have some suggestions and advice.

Please explain what does „the forest is generally poor” mean?

“essential role” versus “capacity is therefore very low” – if something has an essential role, it should be strong, not poor.

Line 40 - Please check million and billion numbering. In the English language, there is another way to write large numbers.

Lines 50-51 – The authors wrote about urbanization and population growth, which are factors influencing floods. Please also include negative human activity (e.g. deforestation) and climate change that is causing droughts and floods in many regions of the world. Urbanization causes entering agricultural areas close to cities, heat islands are formed in cities.The frequency of extreme weather events and their intensity is now more evident. I propose to read the article: Nexus between water, energy, food and climate change as challenges facing the modern global, European and Polish economy. AIMS Geosciences, 6, 397-421 or Exploring interactions in the local water-energy-food nexus (WEF-Nexus) using a simultaneous equations model. Science of The Total Environment, 703, 135034

Line 58 - Water Assessment Tool should be WAT. SAWT is Soil….

Line 358 - Please check if the formulas of Root Mean Square Error is correct written?

In conclusion, please enter two sentences showing how the research is important for the readers.

Author Response

We would like to express our sincere gratitude to the reviewer for the positive and encouraging comments and for the overall appreciation of work. We have carefully read and addressed all comments, point by point, as indicated below. 

Reviewer no. 2

Comment

Response

Reference

Thank you very much for the possibility to read your paper concerning Machine learning and Remote Sensing application for extreme climate evaluation. After reading that paper I have some suggestions and advice.

We would like to express our sincere gratitude to the reviewer for the positive and encouraging comments and for the overall appreciation of work. We have carefully read and addressed all comments, point by point, as indicated below.

Please explain what does, the forest is generally poor” mean?

“essential role” versus “capacity is therefore very low” – if something has an essential role, it should be strong, not poor.

Corrected as suggested.

Lines 143-151

Line 40 - Please check million and billion numbering. In the English language, there is another way to write large numbers.

Corrected as suggested.

Line 45

Lines 50-51 – The authors wrote about urbanization and population growth, which are factors influencing floods. Please also include negative human activity (e.g. deforestation) and climate change that is causing droughts and floods in many regions of the world. Urbanization causes entering agricultural areas close to cities, heat islands are formed in cities. The frequency of extreme weather events and their intensity is now more evident. I propose to read the article: Nexus between water, energy, food and climate change as challenges facing the modern global, European and Polish economy. AIMS Geosciences, 6, 397-421 or Exploring interactions in the local water-energy-food nexus (WEF-Nexus) using a simultaneous equations model. Science of The Total Environment, 703, 135034

Thank you for suggesting these very relevant titles. All were consulted and cited accordingly.

Lines 56-59

Line 58 - Water Assessment Tool should be WAT. SWAT is Soil …..

Corrected as suggested.

Line 66

In conclusion, please enter two sentences showing how the research is important for the readers.

Thank you, we corrected the conclusion section as suggested.

Lines 584-609

Round 2

Reviewer 2 Report

Thank you very much for your answers.